

**Simultaneous observations by sky radiometer and MAX-**
**DOAS for characterization of biomass burning plumes in**
**central Thailand in January-April 2016**
**Hitoshi Irie[1], Hossain Mohammed Syedul Hoque[1], Alessandro Damiani[1], Hiroshi**
**Okamoto[1], Al Mashroor Fatmi[1], Pradeep Khatri[2], Tamio Takamura[1], and**
**Thanawat Jarupongsakul[3]**
[1]{Center for Environmental Remote Sensing, Chiba University, 1-33 Yayoicho, Inage-ku,
Chiba 263-8522, Japan}
[2]{Center for Atmospheric and Oceanic Studies, Graduate School of Science, Tohoku
University, Sendai 980-8578, Japan}
[3]{Department of Geology, Faculty of Science, Chulalongkorn University, Phayathai Road,
Bangkok 10330, Thailand}
**Abstract**
The first intensive multi-component ground-based remote sensing observations by sky
radiometer and Multi-Axis Differential Optical Absorption Spectroscopy (MAX-DOAS) were
performed simultaneously at the SKYNET/Phimai site located in central Thailand (15.18°N,
102.56°E) from January to April 2016. The period corresponds to the dry season associated
with the intense biomass burning (BB) activity around the site. The near-surface concentration
of formaldehyde (HCHO) retrieved from MAX-DOAS was found to be a useful tracer for BB
plumes. As the HCHO concentration tripled from 3 to 9 ppbv, the ratio of gaseous glyoxal to
HCHO concentrations in daytime decreased from ~0.04 to ~0.03, responding presumably to the
increased contribution of volatile organic carbon emissions from BB. In addition, clear
increases in aerosol absorption optical depths (AAODs) retrieved from sky radiometer
observations were seen with the HCHO enhancement. At a HCHO of 9 ppbv, AAOD at a
wavelength of 340 nm reached as high as ~0.15±0.03. The wavelength dependence of AAODs
at 340-870 nm was quantified by the absorption Ångström exponent (AAE), providing evidence



for the presence of brown carbon aerosols at an AAE of 1.5±0.2. Thus, our multi-component
observations around central Thailand are expected to provide unique constraints for
understanding physical/chemical/optical properties of BB plumes.

## 5  1  Introduction

It is well recognized that aerosols contribute the largest uncertainty to the estimate of radiative
forcing (*e.g.*, IPCC, 2013). Biomass burning (BB) is a substantial source of aerosols to the
atmosphere. Black carbon (BC) is a strongly-light-absorbing component of aerosols and can be
emitted in large quantities from BB. In addition, about two-thirds of the global primary organic
aerosol (OA) that should comprise a large amount of ultraviolet (UV)-light-absorbing OA,
known as brown carbon (BrC), originates from BB plumes (Bond et al., 2013). Currently, most
climate models treat OA as purely scattering. Recent laboratory studies suggested that BrC can
enhance net absorption by OA, potentially altering the BB direct radiative forcing from negative
to positive (Kirchstetter et al., 2004; Saleh et al., 2014). Moreover, underestimation in aerosol
absorption over most BB regions was reported by Hammer et al. (2016), who used a global 3-
D chemistry transport model (GEOS-Chem), in which OA is regarded as purely scattering. Thus,
the potential climate effects of BrC aerosols have received considerable attention recently (*e.g.*,
Myhre et al., 2013). In addition, as a result of UV absorption, tropospheric photochemistry can
be significantly affected; GEOS-Chem simulation incorporating UV absorption by BrC showed
a decrease in tropospheric hydroxyl radical (OH) concentration by up to 15% for BB regions,
compared to the simulation without UV absorption by BrC (Hammer et al., 2016). BrC
comprises a wide range of poorly characterized compounds that exhibit highly variable
absorptivity. Assessing the role of BrC in light absorption is further difficult, because BrC is
not only emitted as a primary organic aerosol (POA) but also produced as a secondary organic
aerosol (SOA) through complex formation processes from volatile organic compounds (VOCs)
originating from BB.
This study focuses on the intense BB activity that occurred around central Thailand from
January to April 2016. Characterization for the BB plumes is attempted using our unique remote
sensing observations by the sky radiometer (*e.g.*, Nakajima et al., 1996) and the Multi-Axis
Differential Optical Absorption Spectroscopy (MAX-DOAS) (*e.g.*, Irie et al., 2011) for both
viewpoints of the optical properties of aerosols (aerosol absorption optical depth, AAOD and



absorption Ångström exponent, AAE) and the organic gas concentrations (formaldehyde,
HCHO and glyoxal, CHOCHO) in BB plumes.
**2    Observation**
**2.1 Sky radiometer observation of aerosol optical properties**
We conducted ground-based remote sensing observations using the sky radiometer and the
MAX-DOAS at SKYNET/Phimai site (15.18°N, 102.56°E) located in central Thailand from
January to April 2016. The period corresponds to the dry season with the occurrence of intense
BB around the site. Indeed, satellite data revealed evident enhancements in the carbon
monoxide column concentration and the fire radiative power (FRP) around the
SKYNET/Phimai site in the dry season (Hoque et al., 2018). The sky radiometer (POM-02;
Prede Co., Ltd, Tokyo, Japan), a sun-sky photometer measuring direct and diffuse solar
irradiances, is the main instrument of the international ground-based remote sensing network
SKYNET (*e.g.*, Takamura and Nakajima, 2004; Nakajima et al., 2007). Measurements of the
direct solar and diffuse irradiances within 160º of the center of the Sun were conducted every
10 min. Values of aerosol optical depth (AOD), single scattering albedo (SSA), refractive index
at 340, 380, 400, 500, 675, and 870 nm were retrieved using the Sky Radiometer analysis
package from the Center for Environmental Remote Sensing (SR-CEReS) version 1 (Mok et
al., 2018), in which SKYRAD.pack version 5 (Hashimoto et al., 2012) is implemented to
retrieve aerosol properties, along with all pre- and post-processing programs for the purpose of
the near-real time data delivery. Data at 1020 nm were not used in this study to avoid possible
impact by low AAOD and interferace by water vapor ($H_2O$) on the estimate of the AAE. Cloud
screening was made by the method of Khatri and Takamura (2009) but without using global
irradiance data from a pyranometer.
The SKYNET/sky-radiometer has on-site calibration methods, namely the Improved Langley
(IL) method determing the calibration constant ($F_0$) (*e.g.*, Campanelli et al., 2007) and the Solar
Disk Scan (SDS) method determing the solid view angle (SVA) (*e.g.*, Nakajima et at., 1996;
Uchiyama et al., 2018). Recently, Mok et al. (2018) used retrievals with SR-CEReS to compare
SKYNET/sky-radiometer AOD and SSA data with those derived from a combination of
Aerosol Robotic Netwowk (AERONET), Multifilter Rotating Shadowband Radiometer
(MFRSR), and Pandora observations in Seoul, Korea during and after NASA KORUS-AQ
(Korea U.S.-Air Quality) campaign in 2016 (Mok et al., 2018 and references therein). For most



1 cases, their agreements were found to be within ±0.01 and ±0.05 for AOD and SSA data,

2 respectively, at all wavelengths from 340 and 870 nm, supporting the ability of the on-site

3 calibration methods using IL and SDS.

4 Since the importance of accurate SVA determination was particularly pointed out to better

5 interpret the difference seen in previous SSA comparisons between SKYNET and AERONET

6 (Khatri et al., 2016), sensitivity analysis was made in the present study by conducting additional

7 retrievals using SVAs offset by ±0.01 msr (~±4%), which is likely to correspond to the

8 uncertainty in SVA determined by a single SDS. Both positive and negative offsets were tested

9 but only the positive offset is discussed here, because the negative offset tented to show only

10 little or no impact on SSA, when SSA was close to unity. This is because a smaller SVA leads

11 to a larger SSA (Hashimoto et al., 2012). The impacts by the SVA offset of +0.01 msr on SSAs

12 were estimated to be as small as -0.010±0.005, -0.010±0.005, -0.010±0.005, -0.010±0.006, -

13 0.012±0.007, and -0.011±0.008 at 340, 380, 400, 500, 675, and 870 nm, respectively. Thus,

14 overestimation (underestimation) in SVA leads to underestimation (overestimation) in SSA,

15 but the magnitude was found to be very small at about ±0.01, when the uncertainty in SVA was

16 ~±0.01 msr. The small impact on SSAs should be a result of compensation by the associated

17 change in $F_0$ values; using SVA values offset by +0.01 msr as an input, the IL method employed

18 in SR-CEReS yields smaller $F_0$ values by about 2.1±0.1%, 1.8±0.2%, 1.7±0.2%, 1.2±0.2%,

19 0.7±0.2%, and 0.5±0.1% for 340, 380, 400, 500, 675, and 870 nm, respectively. The resulting

20 smaller $F_0$ leads to a larger SSA (Hashimoto et al., 2012), which is an opposite trend of the

21 direct impact that a larger SVA leads to a smaller SSA (Hashimoto et al., 2012). Results from

22 these sensitivity analyses support the agreement of SSAs within ±0.05 seen in recent

23 comparisons by Mok et al. (2018) during and after NASA KORUS-AQ campaign.

24 Using the AOD and SSA data retrieved, AAOD and AAE values were derived as follows. First,

25 for each measurement and for respective wavelengths from 340 to 870 nm, the AAOD value

26 and its uncertainty ($\varepsilon_{AAOD}$) were calculated with the following equations:

27

28   $$\text{AAOD}(\lambda) = \text{AOD}(\lambda) \cdot [1 - \text{SSA}(\lambda)] \tag{1}$$

29   $$\varepsilon_{\text{AAOD}(\lambda)} = \sqrt{[(1 - \text{SSA}(\lambda)) \cdot \varepsilon_{\text{AOD}}]^2 + (\text{AOD}(\lambda) \cdot \varepsilon_{\text{SSA}})^2} \tag{2}$$

30





For the estimate of $\varepsilon_{AAOD(\lambda)}$, uncertainties for AOD($\lambda$) and SSA($\lambda$) ($\varepsilon_{AOD}$ and $\varepsilon_{SSA}$) were assumed
to be 0.01 and 0.05, respectively, based on comparisons by Mok et al. (2018). Since the
comparisons by Mok et al. (2018) were made using independent measurements having
uncertainties of the same order of magnitude, the actual uncertainties in sky radiometer AOD
and SSA data would be smaller. AAE is calculated as the slope of the linear fit of ln[(AAOD($\lambda$)]
versus ln($\lambda$):
$$\ln[(AAOD(\lambda)] = a - AAE \cdot \ln(\lambda), \qquad\qquad (3)$$
where $a$ is an intercept. This equation is equivalent to expression using the power law:
$$AAOD(\lambda) = K\lambda^{-AAE}, \qquad\qquad (4)$$
where $K$ is a constant. To exclude AAE data associated with large uncertainty, only the data,
which satisfy the criteria that 1) AAOD($\lambda$) exceeds $\varepsilon_{AAOD(\lambda)}$ for all wavelengths and 2) the
correlation coefficient of the linear fit ($R$) is high (-1.0$\leq R \leq$-0.9) are used in the analysis below.
To refine the data set of AOD, SSA, AAOD, and AAE with reduced uncertainty, the daily mean
and its standard deviation with the number of data more than 4 were calculated for 9:00-15:00
LT.

**2.2 MAX-DOAS observation of trace gases**
The MAX-DOAS is an instrument measuring UV-visible spectra of scattered sunlight at several
elevation angles between the horizon and zenith (*e.g.*, Hönninger and Platt, 2002; Hönninger et
al., 2004; Irie et al., 2015). Its measurement is based on the well-established DOAS technique
that quantitatively detects narrow band absorption by trace gases by applying Lambert-Beer
law (*e.g.*, Platt and Stutz, 2008). Since the pioneering study by Hönninger and Platt (2002) and
Hönninger et al. (2004), various types of instruments and algorithms for MAX-DOAS have
been developed worldwide. One of the reasons for this is because ground-based MAX-DOAS
observations at a low elevation angle provide enhanced signals of concentrations of important





trace gases in the boundary layer (*i.e.*, around the instrument altitude) and the concentrations
can be interpreted as being the average over a distance, which is on the same order of or finer
than the horizontal resolution of models and satellite observations but coarser that that of in situ
observations (Irie et al., 2011).
From January to April 2016, our MAX-DOAS system (PREDE, Co., Ltd) (Irie et al., 2011;
Hoque et al., 2018) was operated continuously at the SKYNET/Phimai site together with the
sky radiometer. The spectrometer Maya2000Pro (Ocean Optics, Inc.) was used to record high-
resolution spectra (with the full width at half maximum of around 0.3-0.4 nm) from 310 to 515
nm. Measurements were made at six elevation angles of 2°, 3°, 4°, 6°, 8°, and 70° every 30 min.
MAX-DOAS off-axis elevation angle measurements were limited to below 10° for minimizing
the possible systematic error in oxygen collision complex fitting results but keeping the
measurement sensitivity in the lowest layer of vertical profiles retrieved high (Irie et al., 2015).
Spectral analysis by the so-called DOAS method (Platt and Stutz, 2008) for spectral fitting
using the nonlinear least-squares method and the subsequent vertical profile retrievals using the
optimal estimation method were performed by our retrieval algorithm, JM2 (Japanese MAX-
DOAS profile retrieval algorithm, version 2) (*e.g.*, Irie et al., 2008; Irie et al., 2011; Irie et al.,
2015). Using the recorded high-resolution UV-visible spectra from 310 to 515 nm, the JM2
allows us to retrieve lower-tropospheric vertical profile information for 8 quantities, including
HCHO, CHOCHO, nitrogen dioxide ($NO_2$), and $H_2O$ concentrations, which are analyzed below.
Of vertical profiles retrieved, the present study analyzed data for a layer of 0-1 km, which
corresponds to the lowest layer with the highest sensitivity owing to the longest light path in
profiles retrieved by JM2. The total uncertainties, including random and systematic errors, were
estimated to be 24% (HCHO), 19% (CHOCHO), 15% ($NO_2$), and 18% ($H_2O$) (Irie et al., 2011).
Using the retrieved $H_2O$ concentration, the relative humidity over water ($RH_w$) for the layer 0-
1 km was estimated using NCEP (National Centers for Environmental Prediction) pressure and
temperature data. To be consistent with sky radiometer data, the daily mean values for 9:00-
15:00 LT are analyzed in this study. More detailed descriptions about MAX-DOAS are given
by Irie et al. (2015), Hoque et al. (2018), and references therein.

**3    Results and Discussion**
Time series of multi-components retrieved from sky radiometer and MAX-DOAS observations
at the SKYNET/Phimai site for the intense biomass burning period from January to April 2016



is shown in Fig. 1. As seen in Fig. 1, $RH_w$ for a layer of 0-1 km derived from MAX-DOAS
observations and the surface $RH_w$ from NCEP data confirm that the period was dry around
Phimai, particularly from the beginning of February through the middle of April (from days 32
to 110). For the period of January-April 2016, mean AOD values at 340, 500, and 865 nm were
high at 0.98±0.41, 0.64±0.27, and 0.27±0.11, respectively. The AOD values reached the peak
in late March (around days 80-85), when AAOD values and HCHO, CHOCHO, and $NO_2$
concentrations were synchronously high. From Fig. 1, positive correlations among them are
suggested.
In Fig. 2, CHOCHO concentrations, ratios of CHOCHO to HCHO concentrations ($R_{GF}$), $NO_2$
concentrations, and AAOD values are plotted against the HCHO concentration. As a trace gas
having the longest lifetime in the three potential BB-originating trace gases investigated here
(*i.e.*, HCHO, CHOCHO, and $NO_2$), HCHO was chosen as the standard. We found tight
correlations between CHOCHO and HCHO concentrations. The $R_{GF}$ was suggested to vary
responding to different VOC emissions such as BB and biogenic activities (*e.g.*, Hoque et al.,
2018). As the HCHO concentration tripled from 3 to 9 ppbv, the $R_{GF}$ decreased from ~0.04 to
~0.03 and the $NO_2$ concentration doubled from ~0.6 to ~1.2 ppbv, responding presumably to
the increased contribution of VOC emissions from BB. The $R_{GF}$ values are slightly greater than
those reported by Hoque et al. (2018), whose statistics included data taken in early morning and
late evening, when $R_{GF}$ values were tended to be low compared to mid-day values analyzed in
the present study. At a HCHO concentration of 9 ppbv, AAOD at 340 nm reached as high as
~0.15±0.03. Much larger AAODs were seen at a HCHO concentration higher than 9 ppbv (Fig.
2). These results provide strong observational evidence that aerosols in BB plumes (*i.e.*, POA
and SOA) are absorptive. In addition, Fig. 2 reveals that HCHO is a good tracer for absorption
aerosols from BB.
While BC has been shown to have an AAE of about unity in literature, AAE values greater than
unity are interpreted as BrC (*e.g.*, Kirchstetter et al., 2004). For the whole period from January
to April 2016, the mean AAE was estimated to be 1.57±0.28 for the entire wavelength region
from 340 to 870 nm (Fig. 3). Only for a shorter-wavelength range from 340 to 500 nm, the
mean AAE was estimated to be 1.93±0.59 (Fig. 3). A larger AAE for a shorter-wavelength
range was also reported by Chakrabarty et al. (2010) for BrC in tar balls from smoldering
biomass combustion. Also shown in Fig. 3 are data of the imaginary part of refractive index ($k$)
retrieved from sky radiometer observations, indicating a strong wavelength-dependence. The




wavelength-dependence was quantified similarly to Eq. (3) as the slope ($w$) of the linear fit of
$\ln(k)$ versus $\ln(\lambda)$. The $k$ values in the UV region were as high as 0.01-0.03 but one order of
magnitude smaller than that of BC (~0.71) (Bond and Bergstrom, 2005). Using the
parameterization derived by Saleh et al. (2014) and the $k$ value at 550 nm ($k_{550}$) derived by
interpolation in the present study (~0.012), the BC-to-OA ratio of the emissions from BB
($R_{BC/OA}$) around Phimai was estimated to be 1.9%. A $R_{BC/OA}$ ratio less than 1.9% is suggested
for smoldering combustion of duffs investigated by Chakrabarty et al. (2010) as their $k_{550}$ value
is smaller than that estimated for Phimai in the present study.
Since HCHO is a good tracer for absorption aerosols from BB as mentioned above, it is
interesting to investigate the dependence of AAE on the HCHO concentration. We found,
however, that their correlations were weak and the AAE at a HCHO of 3 ppbv (~1.7) tended to
be higher than the AAE values at higher HCHO concentrations (~1.5) (Fig. 4). Although
uncertainty in the estimate for the single data of daily mean AAE was as large as 0.3-0.5, it can
be seen that AAE tended to converge to 1.5±0.2 at a higher HCHO concentration in BB plumes.
According to smog chamber experiments by Saleh et al. (2014), aerosol absorptivity depends
largely on burn conditions, not fuel type. In addition, the size distribution and the mixing state
of BC and OA can be important factors for AAE (*e.g.*, Russel et al., 2010; Kirchstetter et al.,
2004). It was also reported that non-absorbing shells over BC cores can lead to AAE greater or
less than unity (Gyawali et al., 2009). Despite such a complexity in interpretation of the
variation in AAE and the uncertainty in sky-radiometer-retrieved AAE as large as 0.3-0.5, we
attempted to interpret possible enhancement in AAE at a HCHO of 3 ppbv. For this, using the
parameterization of Salah et al. (2014) and the $k_{550}$ values, the $R_{BC/OC}$ ratio for a low HCHO
case at 3 ppbv was calculated to be 0.013, which was smaller than the $R_{BC/OC}$ ratios at higher
HCHO concentrations (*e.g.*, 0.023 at a HCHO of 9 ppbv) (Fig. 4). A smaller $R_{BC/OC}$ ratio can
be attributed to the lower-temperature BB. In this case, the lower-temperature BB could yield
only small values of AOD, AAOD, HCHO, and CHOCHO (*i.e.*, the magnitude of BB emissions
was weak) but a high value of AAE about 1.7 (*i.e.*, as a results of a smaller $R_{BC/OA}$ ratio for
emissions) (Fig. 4). The other interpretation for the enhancement in AAE at a HCHO of 3 ppbv
is that we observed more photochemically-aged BB plumes at smaller HCHO concentrations.
As the photochemical aging occurred, more SOA should have been produced, leading to
stronger wavelength-dependence of absorption. Considering a large uncertainty in AAE data
used here, further investigation using more data from our multi-component observations by sky



radiometer and MAX-DOAS is encouraged to better interpret the characteristics of BB plumes
observed in this study. In addition, the results presented here are expected to be unique
constraints for understanding physical/chemical/optical properties of BB plumes.
**4   Conclusions**
We conducted ground-based remote sensing observations using the sky radiometer and the
MAX-DOAS at SKYNET/Phimai site in central Thailand from January to April 2016 to
characterize optical properties of aerosols and organic gas concentrations in BB plumes. We
found that the HCHO concentration for a layer of 0-1 km retrieved from MAX-DOAS was a
useful tracer for BB plumes. As the HCHO concentration tripled from 3 to 9 ppbv, the $R_{GF}$
decreased from ~0.04 to ~0.03, in respond presumably to the increased contribution of VOC
emissions from BB. In addition, AAODs increased with HCHO. At a HCHO of 9 ppbv, AAOD
at 340 nm reached as high as ~0.15±0.03. The AAE at 340-870 nm was about 1.5±0.2,
indicating the presence of BrC aerosols. The results from our multi-component observations
around central Thailand are expected to be unique constraints for understanding
physical/chemical/optical properties of BB plumes.

**Acknowledgments**
Support from Mr. Vijak Pangsapa and the Bureau of Royal Rainmaking in Agricultural Aviaion
(BRRAA) is gratefully acknowledged. This work was supported by JSPS KAKENHI Grant
Number JP16K00512, JSPS KAKENHI Grant Number JP15H01728. and JST CREST Grant
Number JPMJCR15K4.





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



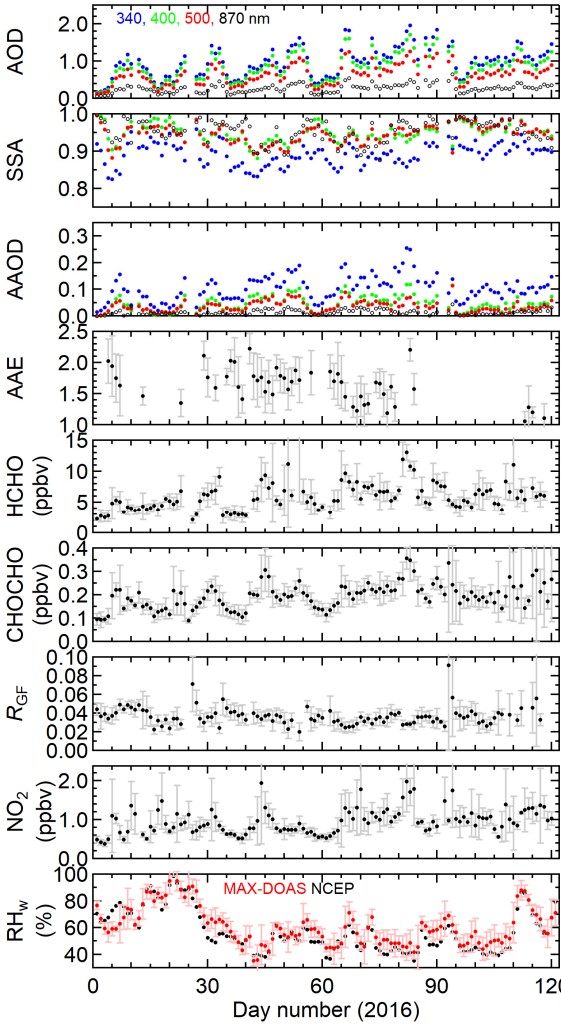

Fig. 1. Time series of multi-components retrieved from sky radiometer and MAX-DOAS

observations at Phimai, Thailand for the intense BB period from January to April 2016. Daily

means for 9:00-15:00 LT are plotted. Their 1σ standard deviations are shown by error bars.

AOD, SSA, and AAOD values for different wavelengths are shown in different colors. For $RH_w$,

red symbols indicate MAX-DOAS-derived $RH_w$ for a layer of 0-1 km and black symbols

indicate the surface $RH_w$ from NCEP data.





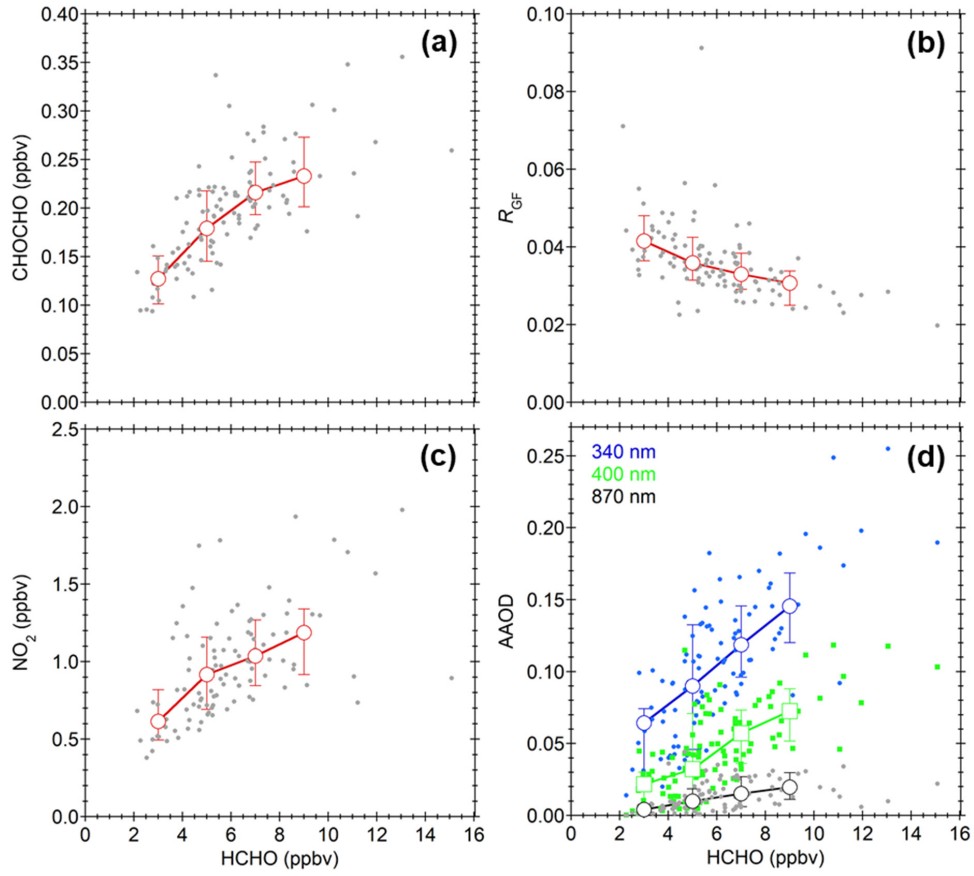

Fig. 2. (a) CHOCHO concentration, (b) $R_{GF}$, (c) $NO_2$ concentration, and (d) AAOD values
plotted as a function of HCHO concentration. AAOD values at 340, 400, and 870 nm are shown
in blue, green, and black, respectively. The medians of respective quantities for each 2-ppbv
bin of HCHO concentration are shown by open symbols. Error bars represent 67%-ranges.



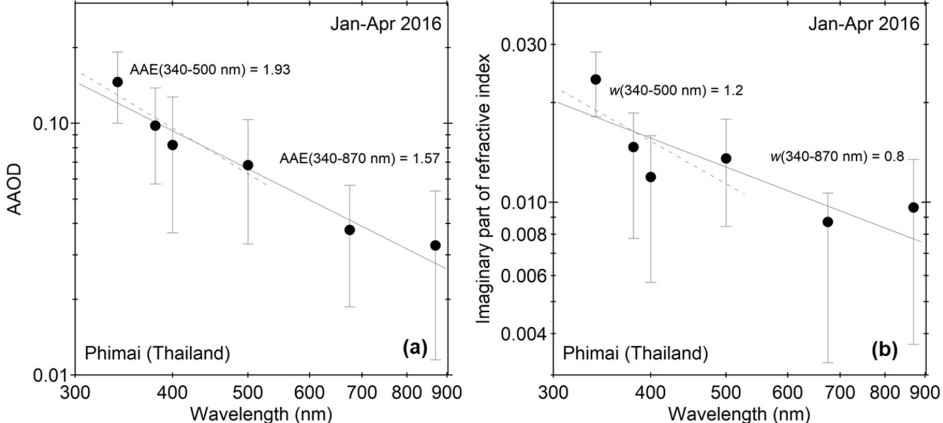

Fig. 3. Spectra of (a) AAOD and (b) imaginary part of refractive index for the period from
January to April 2016. The power law fitting results for 340-870 nm and 340-500 nm are
shown by solid and dashed lines, respectively. Error bars represent 1σ standard deviations for
each wavelength.

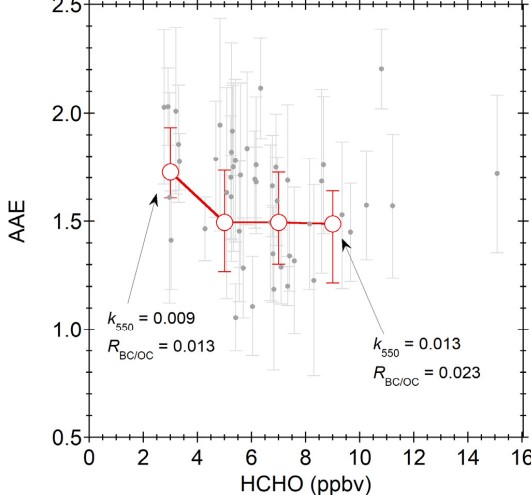

9        Fig. 4. AAE values plotted as a function of HCHO concentration.

