# Peer review of "Simultaneous observations by sky radiometer and MAX- DOAS for characterization of biomass burning plumes in central Thailand in January-April 2016"

_Atmospheric Measurement Techniques, 2018_

## Referee Comment (RC1) · Anonymous Referee #1 · 11 Oct 2018

- General Comments

This manuscript analyzed valuable measurements of trace gases and aerosols including biomass burning events at Phimai, Thailand using Sky-radiometer and MAX-DOAS. However, the descriptions of retrievals and analysis are not clear at least to me. This manuscript is also lack of completeness by depending on other references, even for some essential information. Therefore, I do not fully agree with their major conclusions of this manuscript yet. Hence, I recommend judging this manuscript after major revisions below.

[Figure]

- Major Comments

Pages 5-6 As this manuscript mainly analyzes collocated retrievals of largely varying absorbing aerosols and trace gases, they need to elaborate how they implemented the aerosol properties for trace gas retrievals. According to their prior papers (Irie et al., 2008a, 2011), They fixed aerosol single scattering albedo as 0.95 for all wavelengths, which can propagate nonnegligible systematic biases in the trace gas retrievals. For example, if they used HCHO fitting window as 335-360 nm (Hoque et al., 2018), the SSA values at this wavelength can be differ by up to $\sim$0.15 compared to the Sky-net retrievals (Figure 1). It might have large effect on analysis throughout this manuscript (e.g., Figures 1, 2, 4). If the authors utilized collocated aerosol properties from Sky radiometer for trace gas retrieval, please describe those. If not, at least they need to analyze error estimation of trace gas retrievals due to the biases of SSA between Sky radiometer retrievals and assumption.

Page 7, lines 20 - 24 This paragraph includes one of the main conclusions of this manuscript. They insist that aerosols in BB plumes are absorptive by suggesting high AAOD values at 340 nm. However, monochromatic AAOD is not a straightforward parameter to represent absorption 'property', since it is function of absorption (SSA) and amount (AOD). As the authors already have SSA retrievals, and I don't think AAOD is prior to SSA for the analysis. Therefore, I recommend to additionally focus on spectral SSA retrievals of BB aerosols to clearly show their absorption properties (e.g., like figure 3, for several BB aerosol events).

Page 7, lines 24-25 HCHO is well correlated to the BB in this manuscript. However, it might not be true at different place and time where/when there are other major sources of HCHO (High concentration of HCHO does not mean there is BB event nearby at any place and time). Thus, authors need to carefully state this sentence, which is one of the major conclusions, with specific time and location throughout the manuscript (e.g., BB is the major sources of HCHO at this time and location, with reference if available).
- Specific Comments

Page 3, lines 25-28 Please add radiometric calibration method and accuracy for sky-scan (diffuse sky) measurements of Sky radiometer.

Page 6, lines 7-9 Please describe oversampling, typical SNR for the trace gas retrieval.

Page 6, lines 10-12 Please add more description of information content (e.g., degrees of freedom) of HCHO, CHOCHO, NO2 profile retrievals.

Page 6 Please add table or description of fitting windows, cross section database (including their reference) of each species.

Page 7, lines 9-10 Please insert a sentence that describes why RGF is important for atmospheric chemistry.

Page 7, lines 10-12 This sentence is not clear to me. Do you mean this? : 'HCHO was chosen as a standard tracers of BB, since it has the longest lifetime among the three potential BB-originating trace gases investigated here (i.e., HCHO, CHOCHO, and NO2).' If so, please suggest their typical lifetime (with references) to clarify this sentence.

Page 7, lines 12-13 Please suggest statistics (e.g., correlation coefficient, RMSE)

---

## Referee Comment (RC2) · Anonymous Referee #2 · 11 Dec 2018

The paper compares observations by sky radiometer and MAX-DOAS for characterization of biomass burning plumes in central Thailand in the period January-April 2016. Although similar measurements have already been published it is of scientific value to have this additional data obtained with different techniques. The manuscript is well structured but at some places the explanations could be more precise. The topic of the manuscript is interesting as additional report of the ratio CHOCHO to formaldehyde from a site in central Thailand, where biomass burning seems to play an important role. However, I am not totally convinced that there is a technical contribution, from the

point of view of the measurement methods. This is probably due to the lack of details of the procedures followed, especially regarding the MAX-DOAS observations and profile retrievals.

My first comment concerns the lack of some relevant information: Description of the site, location of the instruments, description of the MAX-DOAS instrument (e.g. is it thermally stabilized? If not, how often was the DC measured?). The MAX-DOAS of the PREDE Co. Ltd. contains a MAYA2000Pro? Please specified the technical details of the spectrometer (e.g. it uses a slit or just an optical fiber?). The measurements were done at  $2^{\circ}$ , $3^{\circ}$ , $4^{\circ}$ , $6^{\circ}$ , $8^{\circ}$ , $70^{\circ}$  elevation angles and the measure at 70° elevation angle was used as reference. Is that correct? Please justified and describe the method. There are also no details about the fitting windows and the cross section used for the trace gas analysis. A table may be useful.

My second comment concerns meteorological information, which is not mentioned in the manuscript. Has NCEP a meteorological station at the site? Since the RH was in January over 60%, was it not necessary to include a scale factor in the DOAS analysis?

My last comment concerns the missing information of the used parameters to retrieve the vertical column of the trace gases: which were the inputs used to retrieve the lower tropospheric vertical profiles? Where do they come from the estimated errors? The site by Irie et al. 2015 is Cabauw, so the parameters are probably different as the parameter used in Phimai.

---

## Author Comment (AC1) · 26 Dec 2018

We thank the reviewer very much for reading our manuscript carefully and giving us valuable comments. Detailed responses to the comments are given below.

*- Major Comments*

*Pages 5-6 As this manuscript mainly analyzes collocated retrievals of largely varying absorbing aerosols and trace gases, they need to elaborate how they implemented the aerosol properties for trace gas retrievals. According to their prior papers (Irie et al.,*

*2008a, 2011), They fixed aerosol single scattering albedo as 0.95 for all wavelengths, which can propagate nonnegligible systematic biases in the trace gas retrievals. For example, if they used HCHO fitting window as 335-360 nm (Hoque et al., 2018), the SSA values at this wavelength can be differ by up to 0.15 compared to the Sky-net retrievals (Figure 1). It might have large effect on analysis throughout this manuscript (e.g., Figures 1, 2, 4). If the authors utilized collocated aerosol properties from Sky radiometer for trace gas retrieval, please describe those. If not, at least they need to analyze error estimation of trace gas retrievals due to the biases of SSA between Sky radiometer retrievals and assumption.*

Reply: We understand the reviewer's concern. Uncertainty in SSA influences the trace gas retrieval through the aerosol retrieval (e.g., Irie et al., 2011). Our detailed error estimation indicates that 1) an influence of uncertainty in SSA of $\pm 0.05$ on the retrieval of AOD is as small as 1% (e.g., Irie et al., 2008) and 2) influences of uncertainty in AOD of 50% and 30% on the retrievals of HCHO and CHOCHO volume mixing ratios in a 0-1 km layer were 16-24% and 11-16%, respectively (e.g., Irie et al., 2011; Hoque et al., 2018a, b). As a result, their combined effect of uncertainty in SSA of 0.15 on HCHO and CHOCHO volume mixing ratios (i.e., the effect from SSA to AOD & the effect from AOD to HCHO and CHOCHO) is less than 1-2%. Thus, the effect that the reviewer concerns is very small. However, we added the following sentence in the revised manuscript; "For HCHO (CHOCHO, $NO_2$, and $H_2O$) retrievals, the systematic error was estimated by conducting additional retrievals as JM2 aerosol retrieval uncertainties of 50% (30%), in which uncertainty due to assuming fixed SSA values should be included (Irie et al., 2008; Irie et al., 2011; Hoque et al., 2018a, b)."

*Page 7, lines 20 - 24 This paragraph includes one of the main conclusions of this manuscript. They insist that aerosols in BB plumes are absorptive by suggesting high AAOD values at 340 nm. However, monochromatic AAOD is not a straightforward parameter to represent absorption 'property', since it is function of absorption (SSA) and amount (AOD). As the authors already have SSA retrievals, and I don't think AAOD is*

*prior to SSA for the analysis. Therefore, I recommend to additionally focus on spectral SSA retrievals of BB aerosols to clearly show their absorption properties (e.g., like figure 3, for several BB aerosol events).*

Reply: We agree with the reviewer. We understand that the word "absorptive" was inappropriate. As we would like to insist here that the aerosols in BB plumes absorb UV radiation significantly (rather than that they are absorptive as its absorption property), we revised the manuscript to state that "These results provide strong observational evidence that aerosols in BB plumes (i.e., POA and SOA) absorb UV radiation significantly".

*Page 7, lines 24-25 HCHO is well correlated to the BB in this manuscript. However, it might not be true at different place and time where/when there are other major sources of HCHO (High concentration of HCHO does not mean there is BB event nearby at any place and time). Thus, authors need to carefully state this sentence, which is one of the major conclusions, with specific time and location throughout the manuscript (e.g.,BB is the major sources of HCHO at this time and location, with reference if available).*

Reply: We appreciate this comment very much. In response to this comment, the revised manuscript now states that "In addition, Fig. 2 reveals that HCHO is a good tracer for absorption aerosols from BB, reflecting that BB caused clear enhancements of both HCHO and absorption aerosols, when BB was the dominant sources of HCHO and absorption aerosols over other sources." Similar revisions were made in abstract and conclusions.

*- Specific Comments*

*Page 3, lines 25-28 Please add radiometric calibration method and accuracy for skyscan (diffuse sky) measurements of Sky radiometer.*

Reply: We also realize that they are important. The radiometric calibration was performed by the Improved Langley method and the Solar Disk Scan method, both of

which have been mentioned already. Their combined uncertainty for AOD and SSA, which are final retrieval products, have also been discussed in section 2.1 already.

*Page 6, lines 7-9 Please describe oversampling, typical SNR for the trace gas retrieval.*

Reply: Following this comment, information on the oversampling has been added in the revised manuscript. We also revised the manuscript to mention a typical SNR for trace gas retrievals as the residual for DOAS fitting, which was usually as low as below 10-3 (corresponding to a SNR of 1,000).

*Page 6, lines 10-12 Please add more description of information content (e.g., degrees of freedom) of HCHO, CHOCHO, $NO_2$ profile retrievals.*

Reply: Following this comment, the manuscript has been revised to state that the degrees of freedom for signal for trace gas vertical profiles retrieved here were usually 1-2.

*Page 6 Please add table or description of fitting windows, cross section database (including their reference) of each species.*

Reply: We used fitting windows and cross section data identical to those described by Irie et al. (2011, 2015) and Hoque et al. (2018a). This is now mentioned in the revised manuscript.

*Page 7, lines 9-10 Please insert a sentence that describes why $R_{GF}$ is important for atmospheric chemistry.*

Reply: Following this comment, a sentence describing the importance of $R_{GF}$ for atmospheric chemistry has been inserted here, as "the $R_{GF}$ is important for atmospheric chemistry as it would vary responding to different VOC emissions such as BB and biogenic activities (e.g., Hoque et al., 2018a, b)."

*Page 7, lines 10-12 This sentence is not clear to me. Do you mean this? : 'HCHO was chosen as a standard tracers of BB, since it has the longest lifetime among the*

*three potential BB-originating trace gases investigated here (i.e., HCHO, CHOCHO, and NO$_2$).' If so, please suggest their typical lifetime (with references) to clarify this sentence.*

Reply: In response to this comment, the sentence has been revised to "HCHO was chosen as a standard, since its lifetime was likely comparable to or longer than the other two potential BB-originating trace gases investigated here (i.e., CHOCHO, and NO$_2$) (e.g., Li et al., 2013) and its variation range was larger than the other two (Figs. 1 and 2).".

*Page 7, lines 12-13 Please suggest statistics (e.g., correlation coefficient, RMSE)*

Reply: Following this comment, a determination coefficient as statistics is now mentioned in the revised manuscript.

---

## Author Comment (AC2) · 26 Dec 2018

We thank the reviewer very much for reading our manuscript carefully and giving us valuable comments. Detailed responses to the comments are given below.

*My first comment concerns the lack of some relevant information: Description of the site, location of the instruments, description of the MAX-DOAS instrument (e.g. is it thermally stabilized? If not, how often was the DC measured?). The MAX-DOAS of the PREDE Co. Ltd. contains a MAYA2000Pro? Please specified the technical details of*

[Figure]

*the spectrometer (e.g.it uses a slit or just an optical fiber?).*

Reply: In response to this comment, a description of the site and the location of the instruments is now given at the beginning of section 2 of the revised manuscript. We combined the MAX-DOAS of the PREDE Co. Ltd. with the Maya2000Pro spectrometer (temperature-controlled, with a slit of 25 $\mu$m). This is also mentioned in the revised manuscript.

*The measurements were done at 2°, 3°, 4°, 6°, 8°, 70° elevation angles and the measure at 70° elevation angle was used as reference. Is that correct? Please justified and describe the method.*

Reply: Yes, that is correct. Instead of 90°, the 70° elevation angle was adopted as reference to reduce a variation range of signals measured at all the elevation angles, while the integration time was kept constant. In the vertical profile retrieval, the elevation angle setting was fully considered in the computation of differential air mass factors (e.g., Irie et al., 2011, 2015). These are now mentioned in the revised manuscript.

*There are also no details about the fitting windows and the cross section used for the trace gas analysis. A table may be useful.*

Reply: We used fitting windows and cross section data identical to those described by Irie et al. (2011, 2015) and Hoque et al. (2018a). This is now mentioned in the revised manuscript.

*My second comment concerns meteorological information, which is not mentioned in the manuscript. Has NCEP a meteorological station at the site? Since the RH was in January over 60to include a scale factor in the DOAS analysis?*

Reply: Meteorological information has been stated in detail in the paper of Hoque et al. (2018a), which is now referenced at the beginning of section 2. For NCEP, more information (reanalysis, 2.5-degree grid, 6-hourly) is now given in the revised manuscript. Since no RH information is needed for our DOAS analysis, we do not think

that any scale factor is needed.

*My last comment concerns the missing information of the used parameters to retrieve the vertical column of the trace gases: which were the inputs used to retrieve the lower tropospheric vertical profiles? Where do they come from the estimated errors? The site by Irie et al. 2015 is Cabauw, so the parameters are probably different as the parameter used in Phimai.*

Reply: The input parameters used for the vertical profile retrievals are the same as those used by Irie et al. (2015) for Cabauw, the Netherlands. The corresponding error estimates have also been done in their work and references therein. This is now stated in the revised manuscript.